# Lightning strikes as a major facilitator of prebiotic phosphorus reduction on early Earth

Benjamin L. Hess [1,2,3 ✉], Sandra Piazolo [2] & Jason Harvey[2]

When hydrated, phosphides such as the mineral schreibersite, $(Fe,Ni)_3P$, allow for the synthesis of important phosphorus-bearing organic compounds. Such phosphides are common accessory minerals in meteorites; consequently, meteorites are proposed to be a main source of prebiotic reactive phosphorus on early Earth. Here, we propose an alternative source for widespread phosphorus reduction, arguing that lightning strikes on early Earth potentially formed 10–1000 kg of phosphide and 100–10,000 kg of phosphite and hypophosphite annually. Therefore, lightning could have been a significant source of prebiotic, reactive phosphorus which would have been concentrated on landmasses in tropical regions. Lightning strikes could likewise provide a continual source of prebiotic reactive phosphorus independent of meteorite flux on other Earth-like planets, potentially facilitating the emergence of terrestrial life indefinitely.

[1] Department of Earth and Planetary Sciences, Yale University, New Haven, CT, USA. [2] School of Earth and Environment, Institute of Geophysics and Tectonics, The University of Leeds, Leeds, UK. [3] Department of Geology and Environmental Science, Wheaton College, Wheaton, IL, USA. ✉email: benjamin.hess@yale.edu

Life on Earth likely originated by 3.5 Ga[1] with carbon isotopic evidence suggesting as early as 3.8–4.1 Ga[2,3]. Phosphorus is one of the key elements for life, involved in biomolecules such as DNA, RNA, phospholipids, and ATP. While terrestrial abiotic phosphorus is essentially ubiquitous on Earth in the oxidised form of phosphate ($PO_4^{3-}$), it is bound in minerals such as apatite, which are effectively insoluble in water[4]. In contrast, reduced phosphorus such as phosphide ($P^0$) in the form of the mineral schreibersite, $(Fe,Ni)_3P$, has been found to be highly reactive[5–7]. When wetted, schreibersite forms hydrous, activated phosphate capable of forming key basic organic molecules, such as glycerol phosphate, nucleosides and phosphocholine[8,9], and intermediate phosphorus species, such as hypophosphite ($H_2PO_2^-$) and phosphite ($HPO_3^{2-}$)[5,10]. While such intermediate phosphorous species would hinder organic reactions, they may still play an important role in the origin of life by efficiently reacting with solar ultraviolet (UV) radiation and dissolved $HS^-$ to form orthophosphate ($PO_4^{3-}$)[11]. Thus, schreibersite is one commonly accepted source of phosphate for the terrestrial prebiotic synthesis of essential organic phosphate molecules[11–13].

Schreibersite is a common accessory mineral within some classes of meteorites[7] and is also found in some highly reduced glasses formed by lightning strikes called fulgurites[14–16]. The Earth likely experienced a monotonic decline in impactors from the moon forming impact at ~4.5 Ga to present[17,18], providing potentially $10^{5–7}$ kg of reduced phosphorus annually throughout the Hadean and early Archean[11]. Consequently, it has generally been assumed that other schreibersite sources are trivial[13].

In this study, we identify abundant accessory schreibersite spherules in a fulgurite formed from clay-rich soil. We propose that under the conditions on early Earth, phosphorus reduction via lightning strikes is a more significant process than previously appreciated, providing a widespread, quiescent source of reduced phosphorus. Further, this presents a mechanism independent of meteorite flux for continually generating prebiotic reactive phosphorus on Earth-like planets, potentially facilitating the emergence of terrestrial life indefinitely.

## Results

**Fulgurite characteristics.** The fulgurite used in this study is a dm-scale type II fulgurite[16] (Fig. 1 and Supplementary Fig. 1), having formed in clay-rich soils in Glen Ellyn, Illinois, USA in 2016. The core of the fulgurite is massive and glassy, whereas the rim is vesicular and frothy (Fig. 2a, b). The studied fulgurite was analysed using Raman spectroscopy, X-ray fluorescence and diffraction (XRF and XRD), electron dispersive spectroscopy (EDS), and electron backscatter diffraction analysis (EBSD; see "Methods"). While the fulgurite structure is predominantly amorphous silica glass (Supplementary Fig. 2), Raman spectra show that the matrix also contains silicon carbide (SiC) and amorphous, graphitic carbon (Fig. 2c). Silicon elemental maps show well-defined silicon dioxide grains (Fig. 2b). EBSD and XRD data show that crystalline quartz is present within the vesicular fulgurite rim, contrasting with amorphous silicon dioxide in the core (Fig. 2b and Supplementary Fig. 2). No high-temperature quartz polymorphs were found. XRD data show that the parent soil contains alpha quartz, albite, muscovite, microcline, and clinochlore (Supplementary Fig. 2). The fulgurite contains metal spherules, many of which have been identified using EDS and EBSD as $Fe_3P$, the iron endmember of the mineral schreibersite, with the remaining spherules being native iron (Fig. 2d–f). The schreibersite spherules within the fulgurite core are incorporated into the matrix and range from 10 to 100 s of microns in diameter (Fig. 2d, f). In contrast, schreibersite spherules in the fulgurite rim line the edge of vesicles and range from a few to tens of microns in diameter (Fig. 2e).

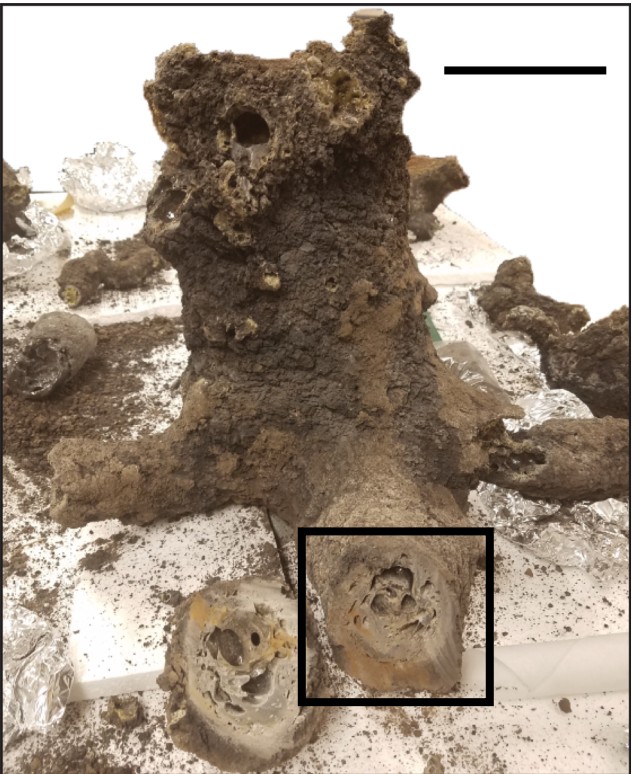

**Fig. 1 Main section of a clay fulgurite created by lightning striking soil.** The thick, glassy interior is coated in burnt soil. The black box indicates the location of the sample analysed in this study (Fig. 2a). Scale bar = 10 cm.

XRF data support the mineral analyses showing a predominance of silicon, aluminium, and iron with minor concentrations of alkali metals (Fig. 3a and Supplementary Table 1). The measured phosphorus concentrations of the fulgurite are lower than the parent soil with the latter being 0.22 wt% $P_2O_5$ and the fulgurite rim and core being 0.11 and 0.16 wt%, respectively (Fig. 3a).

**Formation conditions of the fulgurite.** Lightning strikes heat the impacted material to over 3000 K[16], consistent with the amorphous silica glass in the fulgurite core. The quartz melting isotherm of ~2000 K divides the vesicular rim from the massive core (Fig. 2b); the latter forming from a highly viscous silica melt resulting from the nearly instantaneous melting of the parent soil. The presence of SiC in the fulgurite rim (Fig. 2c) implies a minimum temperature of ~1600 K for the entire sample[19]. Both amorphous silicon dioxide (i.e. former quartz grains) in the core and a lack of high-temperature quartz polymorphs in the rim indicate that the fulgurite core cooled rapidly[20] (Fig. 2b).

The amorphous, graphitic carbon (Fig. 2c) acted as a reducing agent, chemically buffering the system at the graphite-carbon monoxide (CCO) buffer[15], which is ~7 log units below the iron-wüstite (IW) buffer at surface pressure[14]. This is consistent with both the observed native iron spherules (Fig. 2d, e), which form below the IW buffer, and SiC, which forms when oxygen fugacity is at least ~5–7 log units below the IW buffer[14,19]. The inferred fulgurite formation temperatures of >2000 K and highly reducing conditions are consistent with those predicted for the generation of schreibersite[21] and with previous reports of schreibersite in fulgurites[14,16,22].

The difference in phosphorus content between the soil and fulgurite (Fig. 3a) is most likely a result of phosphorus migration from the matrix during the formation of heterogeneously

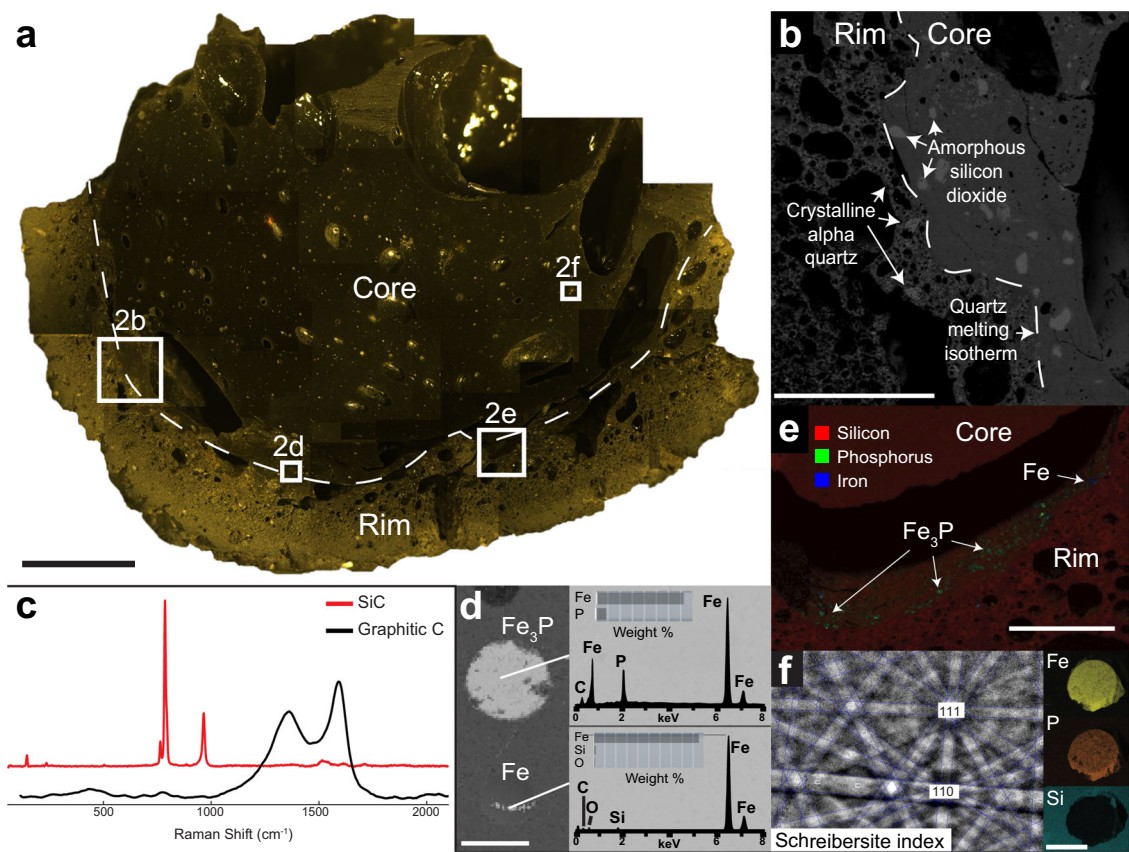

**Fig. 2 Fulgurite sample structure and chemistry. a** Stitched plain light microscope map of the fulgurite marked with key analysed areas. The dashed line signifies the core/rim boundary. Scale bar = 1 cm. **b** SEM silicon intensity map of the contact between the massive, glassy core and vesicular rim. The rim contains crystalline alpha quartz, while the core contains amorphous silicon dioxide. Scale bar = 2.5 mm. **c** Representative silicon carbide (SiC) and amorphous, graphitic carbon Raman spectra found throughout both the fulgurite core and rim. **d** Electron backscatter map with EDS spectra and semi-quantitative measurements of a schreibersite (Fe$_3$P) spherule and several smaller native iron (Fe) spherules lining a vesicle in the fulgurite core. Scale bar = 100 μm. **e** Red-green-blue map of silicon, phosphorus, and iron, respectively, showing Fe$_3$P and Fe spherules lining vesicles in the fulgurite rim. Scale bar = 1 mm. **f** Fe, P, and Si intensity maps of a Fe$_3$P spherule and its electron backscatter diffraction pattern identifying it as the mineral schreibersite. Scale bar = 250 μm.

distributed schreibersite spherules (Fig. 2d–f), resulting in a "nugget effect" during bulk sampling for XRF analysis[23]. By assuming that phosphorus abundance should fall on the equivalent composition line between the soil and fulgurite (Fig. 3a), we can calculate a minimum and maximum estimate of schreibersite formation. The difference between expected P$_2$O$_5$ (0.246 wt%) and observed P$_2$O$_5$ (0.11 and 0.16 wt%) values in the matrix indicates that a minimum of 55% phosphorus was reduced to phosphide in the rim and 35% in the core. Consequently, for the ~25 kg of recovered fulgurite, we estimate 60–172.5 g of schreibersite were formed (see calculation in Supplementary Discussion 1). Thus, under the low redox and high-temperature conditions, phosphorus and iron readily form schreibersite.

Iron is present as a major element in many minerals, while phosphorus is present as a minor element in phosphate minerals such as apatite in most rock types. The reducing agent, graphitic carbon, however, is the primary limiting factor on schreibersite formation in fulgurites. Sufficient carbon is necessary to hold the redox conditions at the CCO buffer, allowing for the formation of highly reduced phases. Figure 3b shows that clay fulgurites are the most likely to be highly reduced and contain schreibersite because they form in soils that are often rich in organic (graphitic) carbon, iron, and phosphorus. In contrast, sand, caliche, and rock fulgurites rarely contain much graphitic carbon[16] and are unlikely to contain schreibersite.

However, even without graphitic carbon, the presence of phosphite (HPO$_3^{2-}$), hypophosphite (H$_2$PO$_2^{-}$)[15], and reduced species of iron oxides[24,25] suggest that many fulgurites are mildly reduced (Fig. 3b). Quartz sand and caliche (graphite-absent) fulgurites have been shown to have 20–70% of their phosphorus reduced to phosphite and hypophosphite compounds[15]. This is consistent with experimental results from electric discharges on phosphate ash[26] and thermodynamic predictions[15,21]. Consequently, most fulgurites will contain some form of reduced phosphorus. Thus, we propose that lightning strikes would generate abundant reduced phosphorus species in terrestrial environments on early Earth.

**Phosphorus reduction by lightning strikes on early Earth.** Hadean zircons indicate that Earth had surface water and continental crust by 4.4 Ga[27], with evidence for liquid water-driven weathering by at least 4.3 Ga[28–31]. The atmosphere was likely composed of H$_2$O, CO$_2$, SO$_2$, and N$_2$[32,33]. Some models suggest most or all of Earth's relatively mafic continental crust formed in the Hadean, undergoing rapid recycling[34–36]. Therefore, Earth was likely habitable with exposed surfaces, largely mafic igneous rocks, by ~4.4 Ga[17]. Recent experiments have shown that komatiites and basalts readily react to form clay minerals and carbonates in CO$_2$- and H$_2$O-rich atmospheres[37,38]. Thus, it is

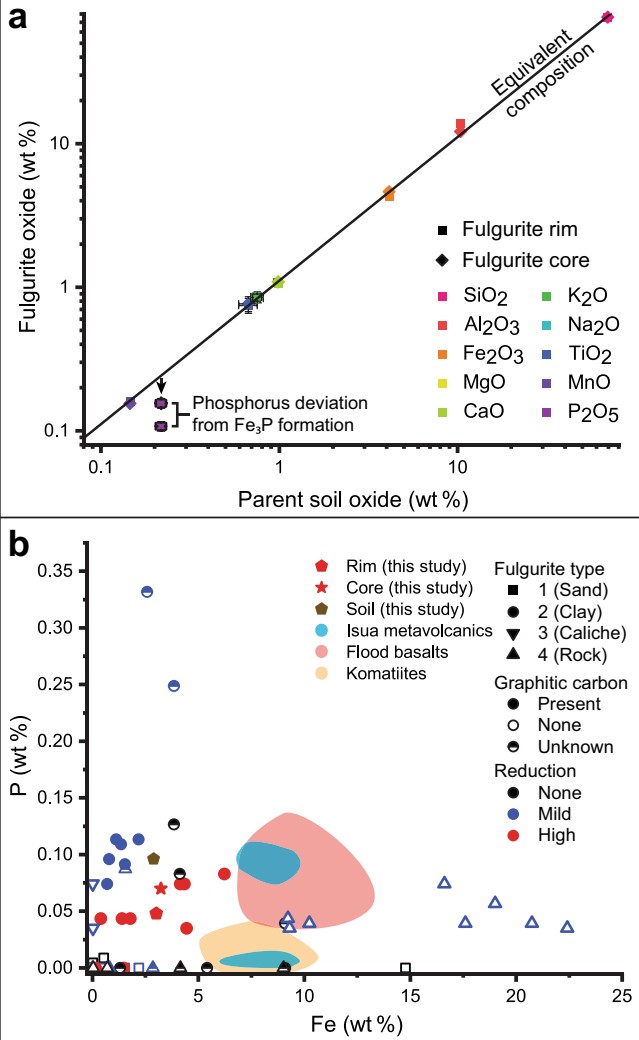

**Fig. 3 Fulgurite chemistry. a** Chemistry of the parent soil vs. fulgurite core and rim in this study. The equivalent composition line is the parent soil normalised to an anhydrous and organic-free composition. Error bars represent per cent uncertainty defined by measurements of standard STSD4 (see 'Methods' and Supplementary Table 1); where not shown, uncertainties are smaller than the used symbol. **b** Iron vs. phosphorus content of fulgurites from this and prior studies[14–16,22,24,25,47,57,58] and three suites of Archean rocks: Abitibi greenstone belt Archean komatiites, Canada[39]; Pilbara continental flood basalts, Australia[40]; Isua supracrustal belt metavolcanics, Greenland[41]. The fulgurites are grouped by type[16], presence of graphitic carbon, and degree of reduction.

likely that mafic rocks with clay and carbonate weathering rinds would have been abundant on Earth's surface during the Hadean and early Archean (Fig. 4a).

The average concentration of iron and phosphorus in a selection of Archean komatiites[39], continental flood basalts[40], and Isua supracrustal belt metavolcanics[41] are consistent with the composition of studied fulgurites that contain mildly to highly reduced phosphorus species (Fig. 3b). In addition, iron-rich carbonates in weathering rinds would partially decompose into graphitic carbon when heated above ~700 K[42,43], a temperature easily attained by lightning strikes (Fig. 4b). Graphitic carbon enhances reduction and, consequently, the formation of phosphite and phosphide (Figs. 2d–f and 4c). All reduced phosphorus species present in fulgurites would be exposed at the surface and able to react with surface water to form dissolved hypophosphite,

phosphite, and phosphate[5,6]. The intermediate phosphorus species readily react with UV radiation and small amounts of HS⁻ dissolved in water to form orthophosphate[11] (Fig. 4d). The HS⁻ needed for this reaction could be sourced from dissolved sulfur species and volcanic gases thought to be abundant on early Earth[12]. Potentially all forms of reduced phosphorus created by lightning strikes could be made available as phosphate for prebiotic chemistry[5,6,11] (Fig. 4d).

Using the described model of early Earth conditions, we estimate the amount of phosphorus reduced by lightning strikes annually on early Earth (Fig. 5; for further details see Supplementary Discussion 2). Our calculations provide an order of magnitude estimate. This means that the output (Fig. 5c) is relatively insensitive to changes in the inputted values (e.g. fulgurite size or phosphorus content) so long as their order of magnitude is reasonably accurate.

First, we determine an annual lightning rate as a function of $pCO_2$. $CO_2$ controls tropospheric temperature and, consequently, the frequency and intensity of storms and associated lightning[44] (Fig. 5a). We estimate the $pCO_2$ throughout the Hadean and early Archean following Kasting[33] (Fig. 5c). We then use the results from an early Earth global circulation model used by Wong et al.[44] to determine a global average lightning rate as a function of $pCO_2$ (Fig. 5c and Supplementary Fig. 3). At present, 75–90% of lightning flashes occur over land[45]. Instead of estimating exposed land area, we choose to assume that a range of 25–75% of lightning strikes occur over land on early Earth. We estimate that 25% of these are cloud-to-ground (i.e. fulgurite-forming) strikes, the same as for present-day Earth[46]. This gives the number of annual fulgurite-forming lightning strikes as a function of time.

To estimate the amount of reduced phosphorus created by each strike, we assume that the average rock fulgurite is 250 g[16,47] with average phosphorus contents between 0.0065 wt% P (komatiites) and 0.044 wt% P (flood basalts; Fig. 3b). We assume that 5–10% of fulgurite-forming strikes are highly reducing and reduce 10–20% of each fulgurite's phosphorus to phosphide. The frequency of extreme reduction is estimated from previous fulgurite studies (Fig. 3b). The per cent of phosphorus reduced to phosphide is a conservative estimate based on the minimum amount reduced in the fulgurite in this study (35–55%) because we assume that there will be less graphitic carbon generated than is available in modern soils. For phosphite and hypophosphite formation, we conservatively estimate that 25–50% of fulgurite-forming strikes are mildly reducing and reduce 25–50% of the fulgurite's phosphorus[15] (Fig. 3b). Consequently, we estimate between 10 and 1000 kg of phosphide and between 100 and 10,000 kg of phosphite and hypophosphite were formed annually in the Hadean and early Archean (Fig. 5c) and were therefore available at the surface for prebiotic organic synthesis.

We note that the above estimates for terrestrial phosphorus reduction are predicated on the validity of an early Earth environment characterised by a significant proportion of exposed landmass and a reactive hydrosphere. These early Earth characteristics are in agreement with early Earth life literature[1,11,48–50]. Consequently, we suggest that so long as such terrestrial early Earth models are valid, our model and resulting estimates are applicable and useful.

## Discussion

Meteorites have been suggested as the primary source of vital prebiotic phosphides for early Earth, enabling the emergence of life[11–13]. To assess the potential importance of the rate of lightning-based phosphorus reduction, we compare it to the estimations of annual meteoritic phosphorus flux from Ritson et al.[11]. We calculate this by beginning with their low and high

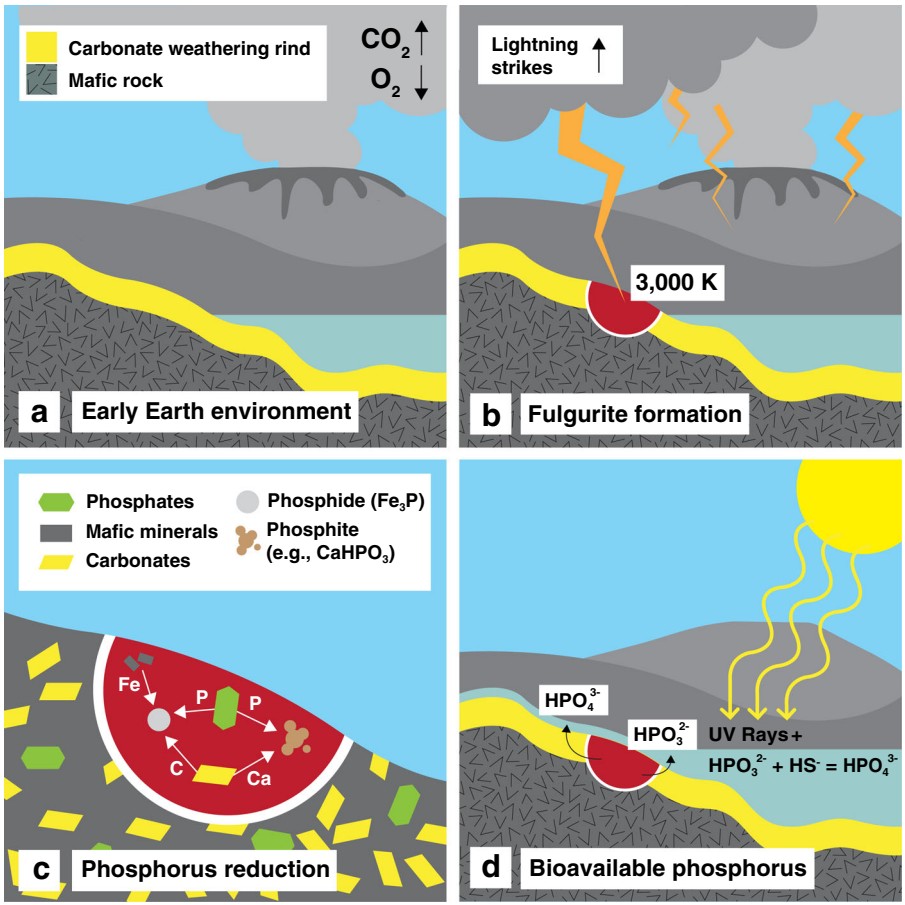

**Fig. 4 Phosphorus reduction by lightning on early Earth. a** Anoxic and $H_2O$- and $CO_2$-rich atmosphere reacts with abundant crustal mafic rocks to form carbonate weathering rinds on the mafic rock. **b** Early Earth had higher lightning rates because of higher $pCO_2$[33,44]. When lightning strikes rock, melting occurs, creating a fulgurite (shown in dark red). **c** Within the fulgurite, iron, phosphorus, and carbon sourced from mafic minerals, phosphates such as apatites, and carbonates, respectively, allow for the formation of reduced phosphides such as $Fe_3P$ and intermediate phosphites such as $CaHPO_3$. **d** Surface water dissolves phosphides and phosphites, which accumulate as hydrous phosphite, hypophosphite, and phosphate species in terrestrial environments. Intermediate phosphorus species react with UV rays and volcanically sourced $HS^-$ to form additional phosphates available for prebiotic chemistry[11].

flux scenarios[17] (Fig. 5c) and then by using the rationale and numbers shown in Fig. 5b.

It is uncertain how much phosphorus would survive an impact in its reduced form. Large impactors, which dominated the mass of the late accretion[17], would either substantially or totally melt or vaporise upon impact[51]. Melts would interact with country rock and solidify at too high a redox state to form schreibersite. Upon vaporisation, it has been argued that schreibersite could precipitate out of the impact plume[52], but given terrestrial redox conditions and plume contamination from vaporised country rock, oxidised phosphorus species may form instead[21]. In addition, atmospheric entrainment, especially of any vaporised water, would further oxidise impact plumes as they cool, potentially preventing schreibersite formation[53] (see Supplementary Discussion 3 for more detail). Therefore, it is not clear what proportion of phosphide would survive. Consequently, we assume that a broad range of 5–50% of phosphorus either survives the impact or re-condenses as phosphide (Fig. 5b, c).

While meteorite flux monotonically decreases through time[17], the rate of lightning strikes remains relatively constant as atmospheric $pCO_2$ reaches a steady state[33,44] (Fig. 5c). Consequently, we estimate that terrestrial reduced phosphorus sourced from lightning strikes surpassed that sourced from meteorites after ~3.5 Ga (Fig. 5c) making phosphorus reduced by lightning strikes significant for a terrestrial emergence of life. Terrestrial environments such as volcanic ponds, terrestrial lakes, tidal pools,

seamounts, and hot springs have been advocated by a number of studies as they allow for important prebiotic compounds to be concentrated in local systems[1,11,48–50] (Fig. 4). Unlike meteorite impacts, which are extremely destructive, lightning strikes would provide a relatively non-destructive, continual source of reactive phosphorus species that would not interfere with the delicate evolutionary steps required for complex prebiotic synthesis[54].

The reduced phosphorus generated by lightning strikes would likely be heterogeneously distributed, being concentrated on tropical landmasses[55] of basaltic compositions (e.g. island arcs and seamounts). Idealised tropical settings would allow for the formation of at least tens to a few hundred grams of reduced phosphorus per $km^2$ per year from lightning strikes.

Our model predicts that there would have been on the order of 1–5 billion lightning flashes per year on early Earth compared to the modern-day value of ~560 million flashes per year[44] (Fig. 5c). Presently, ~100 million cloud-to-ground strikes occur over tropical landmasses with some regions receiving upwards of 100 cloud-to-ground strikes per $km^2$ per year[55]. It is plausible, therefore, that under the elevated lightning rates of early Earth that there may have been at least a few hundred cloud-to-ground (i.e. fulgurite-forming) strikes per $km^2$ per year on some tropical landmasses. In addition, these islands would have been volcanically active, and volcanic plumes from basaltic eruptions can generate lightning, further increasing lightning frequency[56]. Finally, while we propose an average

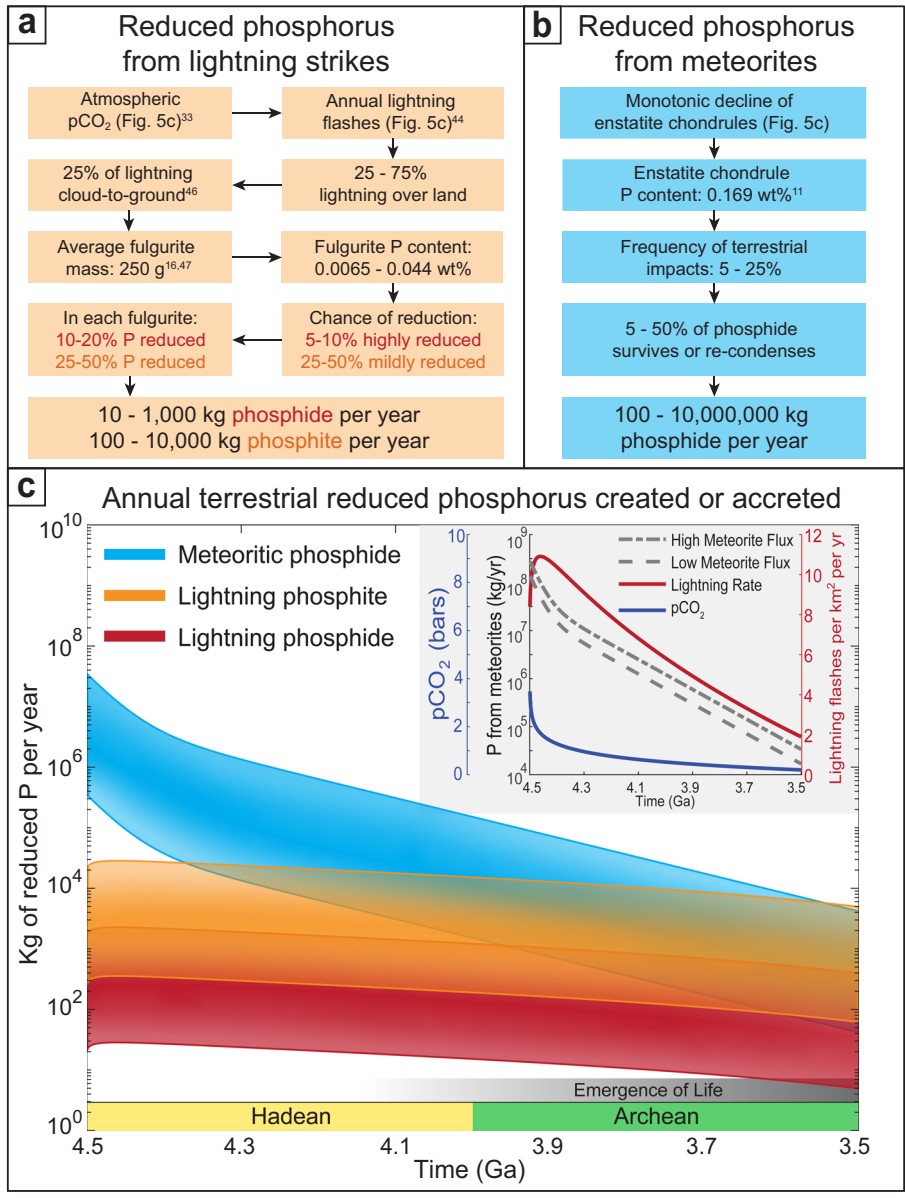

**Fig. 5 Method and calculations for the annual amount of reduced phosphorus sourced from lightning strikes vs. meteorite influx during the Hadean and early Archean. a** Flowchart showing the steps in calculating the annual formation of reduced phosphorus from lightning strikes on early Earth (see main text and Supplementary Discussion 2). **b** Flowchart showing the steps in calculating the annual flux of reduced phosphorus from meteorites on early Earth (see also main text and Supplementary Discussion 3). **c** Mass range of reduced phosphorus produced per year from fulgurite-forming lightning strikes and meteorites as calculated in (**a**, **b**). Inset shows model inputs for atmospheric $pCO_2$ in bars, lightning strikes per $km^2$ per year, and annual total P in kg per year from meteorites[11] as a function of time.

basalt phosphorus content of 0.044 wt% P for our calculations, there certainly would be some basalts with significantly greater phosphorus contents. Therefore, an idealised tropical volcanic island setting could easily generate tens to a few hundred grams of reduced P per $km^2$ per year. As it would take time for the phosphorus to weather out of the fulgurites, the amount of reduced P would build up over long timescales. This would lead to a relatively continuous source of phosphate weathering out of fulgurites in terrestrial environments.

Therefore, lightning may have provided a robust and continual source of terrestrial reduced phosphorus, which could have played a role in the emergence of life. Further, lightning strikes could be an important source of reduced phosphorus on other Earth-like planets. If there is a lightning-rich atmosphere, appropriately exposed lithologies, and an active hydrosphere,

lightning can fulfil the function of in situ phosphorus reduction independent of any meteorite source, potentially indefinitely prolonging the window for the emergence of life on Earth-like planets.

## Methods

**Sample preparation**. The sample analysed in this study (Fig. 2) was cut from the base of the fulgurite (Fig. 1). The fulgurite sample was hand polished with alumina powder prior to conducting Raman spectroscopy. The sample was then polished further with diamond paste prior to scanning electron microscopy (SEM) and EBSD work. At no point was SiC used to polish the sample.

**Raman spectroscopy**. Raman point analysis and mapping were performed using a Horiba LabRAM HR Evolution Raman confocal microscope at the Materials Preparation and Measurement Laboratory, University of Chicago. Data were collected using 532 and 473 nm lasers, ×100 objective, and an Andor EMCCD detector. Spectra were processed using the LabSpec6 software.

**X-ray fluorescence spectroscopy**. Major element abundances were measured by XRF on soil collected adjacent to the fulgurite and two powders collected from a crushed section of each of the fulgurite core and rim. XRF data were collected using a Rigaku ZSX Primus II with a rhodium tube. The major elements were determined on fused glass beads prepared from dried powders with a sample to flux ratio 1:10, 66% Li tetraborate:34% Li metaborate flux. Loss on ignition was determined gravimetrically by measuring the mass difference on aliquots of powdered material both before and after heating to >1000 °C for 1 h. Reproducibility of certified reference material STSD4 analysed alongside the samples was ±≤3 relative % for $SiO_2$, $Al_2O_3$, MnO, CaO, and $K_2O$; 6 to 8 relative % for $Na_2O$, $P_2O_5$ and MgO; 13 relative % for $TiO_2$.

**X-ray diffraction**. Qualitative mineral identification was obtained using a Bruker D8 X-ray diffractometer in the School of Earth and Environment at the University of Leeds. Mineral indexing patterns were interpreted using EVA© software.

**SEM mapping**. SEM-based backscatter electron imaging and EDS mapping were performed using a Tescan VEGA3 XM tungsten source machine at the Leeds Electron Microscopy and Spectroscopy centre (LEMAS), University of Leeds, run at high vacuum, acceleration voltage of 20 kV and working distance of 12 mm. EDS spectra obtained were processed using Oxford Instruments AZtec software. EBSD analyses were performed using an FEI Quanta 650: FEGESEM environmental SEM with Oxford Instruments INCA 350 EDX System and Symmetry EBSD detector, also at LEMAS, University of Leeds. Data acquisition conditions were: 70° tilted sample orientation, 20 kV acceleration voltage, 8.0 nA beam current, high vacuum and 23–25 mm working distance.

## Data availability

The authors declare that the data supporting the findings of this study are available within the paper and its Supplementary information files.

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

## Acknowledgements

We thank the Geology and Environmental Science Department of Wheaton College, IL and the Tuscher family for the use of their fulgurite. We also thank L. Entwisle for help with figure design and R. Walshaw for help with SEM/EBSD analysis. We gratefully acknowledge financial support from Yale University, the MRSEC Shared User Facilities at the University of Chicago (NSF DMR-1420709) and the School of Earth and Environment Student Support for SEM analyses (University of Leeds).

## Author contributions

B.L.H. initiated this project, prepared the sample, collected Raman data, assisted in SEM data collection, processed and interpreted data, and is the primary author of the paper. S.P. collected EBSD data, assisted in data interpretation, and assisted in manuscript layout and revisions. J.H. collected XRF and XRD data, assisted in SEM data collection, assisted in data interpretation, and assisted in manuscript layout and revision.

## Competing interests

The authors declare no competing interests.
