## [Peer Review File · Nature Communications]

REVIEWER COMMENTS

Reviewer #1 (Remarks to the Author):

Phosphate was undoubtedly required for life to emerge on Earth, but the overwhelming majority of Earth's phosphate, obtained from planetary accretion, would have been tied up as insoluble minerals such as apatite. This has raised the age old question of how phosphorous could have been made available for life's beginning. Meteoritic delivery of reduced phosphorus has generally been invoked and these phosphide minerals undergo aqueous alteration typically releasing soluble phosphate, phosphite and hypophosphite - the latter two compounds can be oxidised to phosphate by UV light and hydrogen sulphide. The authors propose that lightning strikes, known to produce reduced phosphorus species in fulgurites, could have provided a substantial ancillary mechanism to meteorites for the supply of soluble phosphorus to subaerial landmasses throughout the Hadean and surpassed phosphorus supply by meteorites in the Archaeon. Whilst the idea of reducing insoluble phosphate minerals to soluble reduced species is not new, this is by far the most rigorous treatment of analytical and modelled data to demonstrate how much reduced phosphorus lightning could have generated (as far as I am aware). One could argue therefore, that this work lacks slightly in novelty, however, I think it is more than outweighed by the interest it will spark among those in the origins of life field and in phosphorus geochemistry on early Earth, as I feel most readers will be surprised that such quantities of phosphide, phosphite and hypophosphite could have potentially been formed due to lightning.

The paper was clearly presented and generally well written barring a couple of minor points. The main one being that the description of phosphate as a reactive or unreactive species will no doubt confuse some readers. Phosphate (by which I assume the authors mean orthophosphate) is generally considered unreactive (as far as chemistry and biology are concerned and this is the implied fate for phosphate here), which is why activating agents or dehydrative conditions are employed to render it reactive. Thus, I would recommend a degree more specificity when discussing phosphate. Such as:

In line 18 – apatite is unreactive, as regards a source of phosphate, because it is effectively insoluble in water. This should be clarified and Gulick (Am. Sci. 1955) should probably be cited also.

In line 20 – 'reactive phosphate' is used, as far as I can tell, to cover what is most likely an activated form of phosphorous in the +5 oxidation state (probably still bound to the metal making it electrophilic) and orthophosphate which is used in a eutectic phase to reduce water activity and allow choline to be phosphorylated. Remove Ref. 9 from line 21 and change 'reactive phosphate' to 'activated phosphate'.

In line 25 – 'key reactive phosphates', it is unclear whether the authors refer to orthophosphate which can be used as a source of phosphate or thiophosphate which is an activated form of phosphate produced in Ref. 10. Also in line 128 - please clarify both.

Line 25/26 – 'commonly accepted source of reactive phosphates' should be changed to 'commonly accepted source of phosphate and activated phosphate' or 'commonly accepted source of phosphate'

Line 131 – 'reactive phosphate', presumably as the previous comment?

I think there is a typo on line 8 – 'hypophosphate' should be 'hypophosphite'

The description and treatment of the data seems fair to have been performed and presented well. It does beg one question though. If we assume 250g for the average fulgurite generated and, at the upper end of the estimate, 0.044 wt% is P of which 50% is reduced, we can expect $(250/100) \times 0.022 = 56$ mg of reduced P per strike. Some discussion of how this translates for utility in prebiotic chemistry and/or nascent life is probably required, although I acknowledge estimates of 'how much was enough' are notoriously difficult. Additionally, Glindemann et al. (Orig. Life Evol. Biosph., 1999) suggested volcanic activity could localise a higher density of lightning thus increasing reduced P production in one area, is this something the authors have considered? Lastly, I think out of pure interest the average reader would be intrigued to know how many lightning strikes could have been

occurring on early Earth under the conditions considered in this paper. This may actually assist the previous point – if we consider 56 mg of reduced P per strike is an average, a small percentage of the strikes would be generating far more reduced P and if the number of strikes per year is extremely high (which it seems to be) there would still be a good number of strikes producing significant localised concentrations of reduced P which would be more conducive for prebiotic chemistry and early life. Although this is implicit, it is worth bringing the reader's attention to it.

Reviewer #2 (Remarks to the Author):

This manuscript offers an intriguing new look at the origin and availability of P on the the early Earth. I think it is a fundamentally sound effort and could be published after some minor revisions addressing the issues listed below:

lines 77-78: How can graphite buffer the system redox? It is just one mineral, which is not sufficient for a buffer. There would have to be a buffer reaction, i.e. graphite + what?

lines 94-95: "The reducing agent of graphitic carbon, however, is the primary limiting factor on schreibersite formation in fulgurites." What does this sentence mean? Here you are describing graphitic carbon in a different way to lines 77-78.

lines 110-118: The very early water/weathering model accepted here is based on highly uncertain concepts (published by the references cited). Please at least consider the consequences of an alternate scenario.

lines 145-146: "To estimate the amount of reduced phosphorus created by each strike, we assume that the average rock fulgurite is 250 grams^{15,42...}". This seems to be a key assumption. What is the basis for this? Everything follows from this assumption.

Reviewer #3 (Remarks to the Author):

I find this manuscript to be well-written, very original and possibly ground-breaking for prebiotic chemistry and exobiology studies. The authors did excellent analytical work and present new simple models to interpret their data and the impact of their interpretation for bioavailable sources of P on the early Earth. The process of lightning is argued to be a robust and continual source for terrestrial reactive prebiotic phosphorus, which can also have broader applications to other planetary and moon surfaces. I think that this manuscript is therefore aimed at the right audience with Nature Communications and will be of broad interest. However, I also think that the authors have to strengthen their observations further in order to conclusively identify carbides and phosphides in their fulgurite. I have read their modelling and their crude estimates make sense, although how realistic these are remains to be further analysed in future papers. Overall I would recommend publication of this manuscript after some minor modifications and additions have been implemented in the manuscript.

For instance, it is great that the authors generate EBSD data on the purported schreibersite, however their Kikuchi lines do not uniquely identify this P mineral as Schreibersite. Indeed, the identified planes (110) and (111) might also be seen in other Fe and P minerals (such as orthorhombic phosphoferrite). To more conclusively demonstrate that the key mineral schreibersite is the one they are looking at, I

would suggest the authors also add at least one EDS spectrum that would show the lack of oxygen and only the presence of Fe and P (and Ni, if present). This is likely easy and already available data since they created EDS maps. Otherwise the presence of phosphide would not be demonstrated unambiguously in my opinion. Strengthening this observation is also critical for the authors' extrapolations into simple models and their broadly-reaching conclusions.

As for evidence for carbides (stated on line 48-50 and in Figure 1), SiC is mentioned, but not demonstrably inside and indigenous in the fulgurite. Considering the porous nature of this rock, I don't see anything convincing (such as petrographic images, special sample preparation methods without SiC, ultrasonication, etc.) that can eliminate the source of SiC from contamination by sample preparation. How was the slab in Fig. 1a polished? Was silicon carbide avoided at every step of sample preparation? It is not uncommon to detect SiC in polished rocks using micro-Raman. This is important to discuss, in order to eliminate possible contamination sources of SiC. I would also suggest that these occurrences be discussed and compared to other reported occurrence of indigenous SiC in sedimentary rocks (or metamorphic sedimentary rocks such as this one), as this was done nicely with schreibersite online 78-81. This is important because this phase is used to extrapolate their new interpretations.

Lines 25-26: This is probably over-stating the importance of Schreibersite for prebiotic chemistry. I agree it is one source of reactive P, but apatite is a much more common rock-forming mineral, which can be made readily bioavailable by simple dissolution under slightly acidic conditions.

Figure S1 : Truly fascinating images of very strange rocks. I would recommend moving Fig. S1a into Fig.1 to help highlight the uniqueness and strange nature of these specimens.

Lines 133-136 and Figure S3 : Are no other variables that influence the surface density of lightning strikes hits than CO₂? I would think it is also highly dependent on relative humidity, which would be good to either specify for this model and/or to model as an additional variable, for broader applicability of the model. In fact, as it is the model does not explain why most lightning occurs in the tropics and/or during stormy weather, which has to be related H₂O levels in air.

Table S1 : What does -0.51% of LOI mean for the fulgurite core? This makes no sense and if it is below detection limit, then write it as such. Also, please also say some more about the STSD4 for reproducibility (e.g. manufacturer or company provider).

Reviewer: Dominic Papineau

Comments on: Lightning strikes as a major facilitator of prebiotic phosphorus reduction on early Earth

In the following, we respond to the comments and suggestions by the editor and reviewers, outlining in detail the changes made. In black – reviewers' comments, in blue – authors' responses.

Reviewer #1:

Phosphate was undoubtedly required for life to emerge on Earth, but the overwhelming majority of Earth's phosphate, obtained from planetary accretion, would have been tied up as insoluble minerals such as apatite. This has raised the age old question of how phosphorous could have been made available for life's beginning. Meteoritic delivery of reduced phosphorus has generally been invoked and these phosphide minerals undergo aqueous alteration typically releasing soluble phosphate, phosphite and hypophosphite - the latter two compounds can be oxidised to phosphate by UV light and hydrogen sulphide. The authors propose that lightning strikes, known to produce reduced phosphorus species in fulgurites, could have provided a substantial ancillary mechanism to meteorites for the supply of soluble phosphorus to subaerial landmasses throughout the Hadean and surpassed phosphorus supply by meteorites in the Archaean. Whilst the idea of reducing insoluble phosphate minerals to soluble reduced species is not new, this is by far the most rigorous treatment of analytical and modelled data to demonstrate how much reduced phosphorus lightning could have generated (as far as I am aware). One could argue therefore, that this work lacks slightly in novelty, however, I think it is more than outweighed by the interest it will spark among those in the origins of life field and in phosphorus geochemistry on early Earth, as I feel most readers will be surprised that such quantities of phosphide, phosphite and hypophosphite could have potentially been formed due to lightning.

We are grateful that Reviewer 1 finds this work of interest, and we also hope it will catalyse further work on this issue.

The paper was clearly presented and generally well written barring a couple of minor points. The main one being that the description of phosphate as a reactive or unreactive species will no doubt confuse some readers. Phosphate (by which I assume the authors mean orthophosphate)

is generally considered unreactive (as far as chemistry and biology are concerned and this is the implied fate for phosphate here), which is why activating agents or dehydrative conditions are employed to render it reactive. Thus, I would recommend a degree more specificity when discussing phosphate. Such as:

In line 18 – apatite is unreactive, as regards a source of phosphate, because it is effectively insoluble in water. This should be clarified and Gulick (Am. Sci. 1955) should probably be cited also.

We have now clarified that apatite is unreactive because it is insoluble and cited Gulick¹ to support this (lines 18-20)

In line 20 – ‘reactive phosphate’ is used, as far as I can tell, to cover what is most likely an activated form of phosphorous in the +5 oxidation state (probably still bound to the metal making it electrophilic) and orthophosphate which is used in a eutectic phase to reduce water activity and allow choline to be phosphorylated. Remove Ref. 9 from line 21 and change ‘reactive phosphate’ to ‘activated phosphate’.

Removed the reference and made the suggested change.

In line 25 – ‘key reactive phosphates’, it is unclear whether the authors refer to orthophosphate which can be used as a source of phosphate or thiophosphate which is an activated form of phosphate produced in Ref. 10. Also in line 128 - please clarify both.

Changed to ‘orthophosphate’ in both cases.

Line 25/26 – ‘commonly accepted source of reactive phosphates’ should be changed to ‘commonly accepted source of phosphate and activated phosphate’ or ‘commonly accepted source of phosphate’

Changed to the latter.

Line 131 – ‘reactive phosphate’, presumably as the previous comment?

Changed accordingly to just phosphate.

I think there is a typo on line 8 – ‘hypophosphate’ should be ‘hypophosphite’

This was indeed a typo, our apologies. Changed.

The description and treatment of the data seems fair to have been performed and presented well. It does beg one question though. If we assume 250g for the average fulgurite generated and, at the upper end of the estimate, 0.044 wt% is P of which 50% is reduced, we can expect $(250/100) \times 0.022 = 56$ mg of reduced P per strike. Some discussion of how this translates for utility in prebiotic chemistry and/or nascent life is probably required, although I acknowledge estimates of 'how much was enough' are notoriously difficult. Additionally, Glindemann et al. (Orig. Life Evol. Biosph., 1999) suggested volcanic activity could localise a higher density of lightning thus increasing reduced P production in one area, is this something the authors have considered? Lastly, I think out of pure interest the average reader would be intrigued to know how many lightning strikes could have been occurring on early Earth under the conditions considered in this paper. This may actually assist the previous point – if we consider 56 mg of reduced P per strike is an average, a small percentage of the strikes would be generating far more reduced P and if the number of strikes per year is extremely high (which it seems to be) there would still be a good number of strikes producing significant localised concentrations of reduced P which would be more conducive for prebiotic chemistry and early life. Although this is implicit, it is worth bringing the reader's attention to it.

We thank the reviewer for these comments. We agree that the reviewer raises an important point. A clear *advantage* of a 'meteoritic' source of reduced P is that meteorites, though rarer and destructive, provide a substantial amount of reduced P at once. A lightning-driven mechanism may generate a comparable amount, but if it is uniformly spread across the Earth's surface, it may not be sufficiently concentrated. Hence, we agree that a discussion of the potential concentration of reduced P is important. We have updated our discussion to offer a more in-depth assessment of the possibility of higher concentrations of phosphorus (lines 202-220) along the following lines:

The distribution of lightning strikes and subsequent reduced P will be concentrated in certain regions. As noted in our manuscript, the vast majority of strikes occur over land. Beyond that, tropical landmasses disproportionately receive over 100 million cloud-to-ground strikes annually with some places receiving an upwards of 100 strikes per km² per year². Our model predicts 1 to 5 billion lightning flashes per year compared to modern Earth's approximately 560 million annual flashes (a result that is now included in lines 206-208 as suggested). It seems reasonable then that some tropical settings may

likewise proportionally experience several hundred cloud-to-ground lightning strikes per km² per year. Additionally, as Reviewer 1 suggests, many volcanic centres have lightning associated with them, including the basaltic eruptions that likely would have been dominant on early Earth³. Lastly, though we have selected an average bulk rock P content, the variation in basaltic compositions allows for the possibility of basalts with 2-3+ times higher P contents.

Consequently, under these idealised conditions there could reasonably be tens to hundreds of grams of reduced P being formed annually per km². This annual production could potentially build up over the years as the fulgurites take time to weather out. Large quantities of reduced P could then constantly be reacting with rainwater and concentrating in terrestrial ponds. Thus, the global estimates we make may be locally concentrated in certain regions, providing significant concentrations of reduced P.

To Reviewer 1's additional comment "*volcanic activity could localise a higher density of lightning thus increasing reduced P production in one area, is this something the authors have considered?*":

We did initially consider the contribution of volcanic lightning in our estimates. However, our estimates are largely an order of magnitude analysis, and from our literature review we found it unlikely that volcanic lightning contributions would affect global lightning rates by anywhere near an order of magnitude. Consequently, we left it out of the calculations for simplicity. But as discussed above, we now let it enter into our further discussion of the possibility of phosphorus concentration as volcanic lightning could be locally important to the discussion of phosphorus concentration (lines 212-214).

Reviewer #2:

This manuscript offers an intriguing new look at the origin and availability of P on the the early Earth.

We are happy to learn that the reviewer agrees with us that this work offers a new look at the origin and availability P on early Earth.

I think it is a fundamentally sound effort and could be published after some minor revisions addressing the issues listed below:

lines 77-78: How can graphite buffer the system redox? It is just one mineral, which is not sufficient for a buffer. There would have to be a buffer reaction, i.e. graphite + what?

Thank you for pointing this out this oversight. The graphite would be part of the graphite-CO (CCO) buffer. At 1 atm pressure and the temperatures inferred for the fulgurite (~2000 K), the CCO buffer falls ~7 log units below the iron-wüstite (IW) buffer⁴. This is consistent with the redox conditions assumed from the analysis of SiC and Fe₃P formation. We have changed lines 81-83 to correctly reference the buffer rather than a single mineral.

lines 94-95: "The reducing agent of graphitic carbon, however, is the primary limiting factor on schreibersite formation in fulgurites." What does this sentence mean? Here you are describing graphitic carbon in a different way to lines 77-78.

The point we are aiming to make here is that in the CCO buffer (see above) graphite is the limiting buffering agent. When the system no longer has graphite, the fO_2 is no longer locally held at the CCO buffer. Then the redox conditions become too high to form reduced phases such as schreibersite and SiC. Intermediate phosphorus species, however, would still be formed. We have rephrased these lines of text to reflect this logic (lines 102-104).

lines 110-118: The very early water/weathering model accepted here is based on highly uncertain concepts (published by the references cited). Please at least consider the consequences of an alternate scenario.

To the extent that there was exposed landmass with an active hydrosphere, our model is applicable as the dominant lithology is reasonably certain. If this were not the case, our model would not be applicable, at least to early Earth. However, this also means that several other models for a terrestrial origin of life would also break down, perhaps favouring an oceanic origin. Consequently, we have added an acknowledgement in our discussion (lines 168-172) that our model is valid and useful only insofar as our assumptions concerning Earth's hydrosphere are valid.

lines 145-146: "To estimate the amount of reduced phosphorus created by each strike, we assume that the average rock fulgurite is 250 grams^{15,42...}". This seems to be a key assumption. What is the basis for this? Everything follows from this assumption.

We agree this is an important concern. The basis for this number is from other fulgurite studies. Primarily our estimate comes from the cited source of Elmi et al.⁵. They study

the mineralogy and composition of a fulgurite formed in granite. They estimate that the fulgurite mass formed per lightning strike “ranges from a few hundred grams to about 30 kg.” The “30 kg” endmember seems consistent with a fulgurite formed in soil such as the one used in our study. However, given the high melting point, specific heat, and low conductivity of minerals compared to soil, it is unlikely that an igneous rock fulgurite would reach this size. However, Elmi et al.⁵ provide a few hundred grams as a minimum mass in a study of a fulgurite formed in granite (i.e., their fulgurite must be at least that size). This suggests to us that 100-1000 grams is a safe order of magnitude range of the sizes for fulgurites formed from igneous rocks intermixed with clay and carbonates as proposed in our manuscript. Thus, 250 grams seemed like a reasonably representative number. We also cite Pasek et al.⁶ which is a paper on the morphology of fulgurites. It does not provide weight ranges, but based on the description, a few hundred grams seems reasonable. We have now added this more explicit discussion of the chosen range of values to Supplementary Discussion 2. Specifically, we now state:

First, we assume that the average rock mass effected by a lightning strike will be 250 grams. This estimate is based on Elmi et al. who studied a granite rock fulgurite. They estimate that the fulgurite mass formed per lightning strike “ranges from a few hundred grams to about 30 kg.” Given the high melting point, specific heat, and low conductivity of igneous minerals, it is likely that fulgurites developed in an igneous rock will fall on the low side of this range. In contrast, in substrates such as soil, fulgurites may reach masses of about 30 kg (Supplementary Fig. 1). Consequently, an approximate mass of ~250 grams is reasonable for a fulgurite generated from lightning striking a carbonate and clay-rich igneous rock.

We would like to stress, ultimately, our calculations presented in Figure 5 cover a broad range and largely depend on the order of magnitude rather than specific numbers. This is now stated at the beginning of the discussion of our model (lines 142-144). We have also added this to Supplementary Discussion 2:

It should be noted that so long as the order of magnitude (10^{2-3}) is correct, our results do not change significantly.

Therefore, our conclusions would not be significantly altered as long as we have the order of magnitude correct, which we believe we do.

Reviewer #3:

I find this manuscript to be well-written, very original and possibly ground-breaking for prebiotic chemistry and exobiology studies. The authors did excellent analytical work and present new simple models to interpret their data and the impact of their interpretation for bioavailable sources of P on the early Earth. The process of lightning is argued to be a robust and continual source for terrestrial reactive prebiotic phosphorus, which can also have broader applications to other planetary and moon surfaces. I think that this manuscript is therefore aimed at the right audience with Nature Communications and will be of broad interest.

We are glad to learn that the reviewer agrees with us that our contribution is well placed in Nature Communications and is “original and possibly ground-breaking”.

However, I also think that the authors have to strengthen their observations further in order to conclusively identify carbides and phosphides in their fulgurite. I have read their modelling and their crude estimates make sense, although how realistic these are remains to be further analysed in future papers. Overall I would recommend publication of this manuscript after some minor modifications and additions have been implemented in the manuscript.

We are delighted that Reviewer 3 is in favour of our work! Please see below how we have addressed the suggestion to further strengthen our approach. We now provide the suggested required extra evidence necessary to further substantiate our results. More details are given below.

For instance, it is great that the authors generate EBSD data on the purported schreibersite, however their Kikuchi lines do not uniquely identify this P mineral as Schreibersite. Indeed, the identified planes (110) and (111) might also be seen in other Fe and P minerals (such as orthorhombic phosphoferrite). To more conclusively demonstrate that the key mineral schreibersite is the one they are looking at, I would suggest the authors also add at least one EDS spectrum that would show the lack of oxygen and only the presence of Fe and P (and Ni, if present). This is likely easy and already available data since they created EDS maps. Otherwise the presence of phosphide would not be demonstrated unambiguously in my opinion. Strengthening this observation is also critical for the authors' extrapolations into simple models and their broadly-reaching conclusions.

We have indeed considered the possibility of phosphoferrite, however two lines of evidence show us that it is indeed schreibersite. The match of theoretical and actual Kikuchi lines is markedly better for Schreibersite than phosphoferrite. However, we agree that EDS point spectra are a useful tool to make sure that only Fe and P are present. So, we have now incorporated an example EDS spectrum and associated Fe/P ratio in our Figure 2. Since no oxygen is present in any of the EDS spectra we measure for the Fe_3P , it cannot be phosphoferrite, and the semi-quantitative Fe/P ratios we obtain work out to approximate the Fe_3P formula.

As for evidence for carbides (stated on line 48-50 and in Figure 1), SiC is mentioned, but not demonstrably inside and indigenous in the fulgurite. Considering the porous nature of this rock, I don't see anything convincing (such as petrographic images, special sample preparation methods without SiC, ultrasonication, etc.) that can eliminate the source of SiC from contamination by sample preparation. How was the slab in Fig. 1a polished? Was silicon carbide avoided at every step of sample preparation? It is not uncommon to detect SiC in polished rocks using micro-Raman. This is important to discuss, in order to eliminate possible contamination sources of SiC. I would also suggest that these occurrences be discussed and compared to other reported occurrence of indigenous SiC in sedimentary rocks (or metamorphic sedimentary rocks such as this one), as this was done nicely with schreibersite online 78-81. This is important because this phase is used to extrapolate their new interpretations.

Thank you for pointing this out as sample contamination can be an issue in some cases. At no point in the process was SiC used in sample prep. We have added the following polishing information to our methods section: "The fulgurite sample was hand polished with alumina powder prior to conducting Raman spectroscopy. The sample was then polished further with diamond paste prior to SEM and EBSD work. At no point was SiC used to polish the sample."

We were unable to resolve SiC Kikuchi patterns in the fulgurite matrix which includes the SiC. Given the rapid timescale of heating and cooling, we believe the SiC is nano- to micro-crystalline and not discernible by conventional EBSD from the amorphous matrix. This is supported by the Raman maps which show occasional SiC spectra points distributed throughout the matrix. There is no spatial relationship between SiC and vugs/vesicles as would be expected if it were introduced via sample preparation.

Additionally, as noted now explicitly in the text (lines 81-85), the redox conditions of SiC formation (5-7 log units below the IW buffer) is consistent with the redox conditions of the graphite-carbon monoxide (CCO) redox buffer (~7 log units below the IW buffer) which we infer to be operating. Finally, there is another fulgurite that has been reported to contain SiC⁷, and other fulgurite studies that have reported similarly ultra-reduced phases such native Si⁴ and FeSi⁶.

Based on the above, there is no reason to doubt the in-situ formation of the observed SiC. Furthermore, the presence of SiC is consistent with the presence of other reduced phases (e.g. Fe, C, and Fe₃P) and the redox conditions inferred in other studies^{4,6,7}. We would also like to note that the presence of SiC is not critical to our analysis which is why we haven't focused more on it in the manuscript itself.

In summary, we have now modified the text to (a) provided the extra information on the polishing procedure (lines 232-234), (b) highlight the lack of spatial correlation of SiC with vugs (lines 54-55), and (c) discuss the redox conditions (lines 81-85). However, we note that if desired we can provide further example Raman spectra and maps to the supplementary information. We ask the editor to provide us with feedback please.

Lines 25-26: This is probably over-stating the importance of Schreibersite for prebiotic chemistry. I agree it is one source of reactive P, but apatite is a much more common rock-forming mineral, which can be made readily bioavailable by simple dissolution under slightly acidic conditions.

We take the point made by the reviewer and have therefore changed the sentence from "schreibersite is the commonly accepted source" to "schreibersite is one commonly accepted source."

Figure S1 : Truly fascinating images of very strange rocks. I would recommend moving Fig. S1a into Fig.1 to help highlight the uniqueness and strange nature of these specimens.

We are appreciative of this suggestion. We have added Supplementary Fig. 1a as its own figure in the main paper; adding it to former Figure 1 would just clutter the figure too much. Supplementary Figs. 1b and 1c remain as supplementary materials for the interested reader.

Lines 133-136 and Figure S3 : Are no other variables that influence the surface density of lightning strikes hits than CO₂? I would think it is also highly dependent on relative humidity, which would be good to either specify for this model and/or to model as an additional variable, for broader applicability of the model. In fact, as it is the model does not explain why most lightning occurs in the tropics and/or during stormy weather, which has to be related H₂O levels in air.

There certainly are many variables that influence the frequency of lightning strikes. However, 1) these are implicit in the data we use, and 2) we are interested in a global average value for our calculations rather than regional variations.

We use the relationship between pCO₂ and average global lightning rates that Wong et al.⁸ produce from their use of the generic LMDZ 3D global circulation model (GCM). This GCM has been used in studies of other planetary scenarios, including Archean Earth⁹, early Mars¹⁰, and terrestrial-mass exoplanets¹¹. This model incorporates the many other factors that influence atmosphere dynamics and subsequent lightning rates such as “robust convection schemes in the lower atmosphere, volatile condensation in the atmosphere and surface, a 2-layer dynamic ocean, and surface and subsurface thermal balance”⁸.

The GCM, therefore, includes many other factors (including atmospheric water content). The study of Wong et al.⁸ uses the GCM to see how variation in pCO₂ affects global average lightning rates. As with modern climate change, pCO₂ acts as a sort of temperature dial because higher pCO₂ warms the air, allowing it to hold more water vapor, creating a feedback loop that leads to global warming. Consequently, pCO₂ is commonly used as a proxy for global temperature. Wong et al. change the pCO₂ content (loosely, the global temperature) and then run the model to determine the average global lightning rates. The rates certainly vary regionally, but they, like us, are primarily interested in the average number of lightning strikes annually. They use this information to calculate an NO budget for early Earth. We use the information to calculate a reduced phosphorus budget. We have modified our Supplementary Discussion 2 to more specifically reflect this relation. The text now reads to clarify the issues noted above:

To estimate the number of fulgurite-forming lightning strikes per year, we use results from Wong et al., who use the generic LMDZ 3D global circulation model (GCM) to estimate average lightning rates on early Earth as a function of pCO₂

in the atmosphere. The rationale behind this is that $p\text{CO}_2$ is a major control on mean surface temperature which in turns controls storm frequency, intensity, and consequently lightning in the GCM. Consequently, $p\text{CO}_2$ can be used to estimate the average global lightning frequency to first order.

It is certainly correct to say that these lightning strikes would have been concentrated over any landmasses near the equator, meaning the global supply would not be homogeneously spread across Earth's surface. However, there is currently no data for constraining the position of continents in the Hadean and early Archean. Therefore, we believe it is fair to only offer a rough global budget. We can use $p\text{CO}_2$ as our only input to estimate approximate average global lightning rates and subsequent phosphorus reduction because the GCM implicit factors in other variables.

We acknowledge that more in depth argumentation along the lines of the above will help the reader to understand and appreciate the choices we have made. Therefore, we have added a discussion about the potential for the regional concentration of reduced P in addressing both Reviewer 1 and Reviewer 3's comments. This new discussion is provided in lines 202-220.

Table S1 : What does -0.51% of LOI mean for the fulgurite core? This makes no sense and if it is below detection limit, then write it as such. Also, please also say some more about the STSD4 for reproducibility (e.g. manufacturer or company provider).

Thank you for pointing this out as it is certainly something we ought to have addressed. A negative LOI actually indicates mass gained upon ignition through oxidation. The fulgurite sample contains abundant highly reduced phases as discussed in the paper (Fe, Fe_3P , SiC). Upon heating for XRF sample prep to approximately 1000 °C, the reduced phases will have oxidised, adding weight through the gained oxygen. One co-author notes that they often see a negative LOI in peridotites that oxidise from Fe^{2+} to Fe^{3+} .

We also added more information about the standard used (STSD4). It is a Canadian certified standard created from stream sediment as part of the Canadian Certified Reference Materials Project (CCRMP) by the company CANMET Mining and Minerals Science Laboratories.

The caption for Supplementary Table 1 now reads:

X-ray Fluorescence (XRF) results for the parent soil, fulgurite rim, and core. The negative loss on ignition (LOI) for the fulgurite core indicates weight gained by the oxidation of highly reduced phases such as Fe (Fig. 1d). A Canadian certified standard stream sediment (STSD4) with a similar composition to the fulgurite was measured along with the samples. The uncertainty is determined by calculating the percent difference between the certified value and the measured standard value. STSD4 was created as part of the Canadian Certified Reference Materials Project (CCRMP) by the company CANMET Mining and Minerals Sciences Laboratories.

References

1. Gulick, A. Phosphorus as a factor in the origin of life. *Am. Sci.* **43**, 479-489 (1955).
2. Gora, E. M. et al. Pantropical geography of lightning-caused disturbance and its implications for tropical forests. *Glob. Change Biol.* **26**, 5017-5026 (2020).
3. McNutt, S. R. & Williams, E. R. Volcanic lightning: global observations and constraints on source mechanisms. *Bull. Volcanol.* **72**, 1153-1167 (2010).
4. Essene, E. J. & Fisher, D. C. Lightning strike fusion: extreme reduction and metal-silicate liquid immiscibility. *Science* **234**, 189-193 (1986).
5. Elmi, C., Chen, J., Goldsby, D. & Giere, R. Mineralogical and compositional features of rock fulgurites: a record of lightning effects on granite. *Am. Mineral.* **102**, 1470-1481 (2017).
6. Pasek, M. A., Block, K. & Pasek, V. Fulgurite morphology: a classification scheme and clues to formation. *Contrib. Mineral Petrol.* **164**, 477-492 (2012).
7. Plyashkevich, A. A., Minyuk, P. S., Subbotnikova, T. V. & Alshevsky, A. V. Newly formed minerals of the Fe-P-S system in Kolyma fulgurite. *Doklady Earth Sci.* **467**, 380-383 (2016).
8. Wong, M. L., Charnay, B. D., Gao, P., Yung, Y. L. & Russell, M. J. Nitrogen oxides in early Earth's atmosphere as electron acceptors for life's emergence. *Astrobiology* **17**, 975-983 (2017).
9. Charnay, B. et al. Exploring the faint young Sun problem and the possible climates of the Archean Earth with a 3-D GCM. *J. Geophys. Res. Atmos.* **118**, 10414-10431 (2013).
10. Forget, F. et al. 3D modelling of the early martian climate under a denser CO₂ atmosphere: Temperatures and CO₂ ice clouds. *Icarus* **222**, 81-99 (2013).

11. Wordsworth, R. D. et al. Gliese 581d is the first discovered terrestrial-mass exoplanet in the habitable zone. *Astrophys. J. Lett.* **733**, L48 (2011).

REVIEWERS' COMMENTS

Reviewer #1 (Remarks to the Author):

The authors have addressed the points I raised and made the necessary clarifications, this manuscript has my support for publication.

Reviewer #3 (Remarks to the Author):

Good job! The authors have responded to my comments in a careful and considerate manner, their contribution is novel and significant, and the modifications implemented, including for the other reviewers' comments are adequate.

In my opinion, this manuscript is ready for publication.

Comments on: Lightning strikes as a major facilitator of prebiotic phosphorus reduction on early Earth

In the following, we respond to the comments and suggestions by the editor and reviewers, outlining in detail the changes made. In black – reviewers' comments, in blue – authors' responses.

Reviewer #1 (Remarks to the Author):

The authors have addressed the points I raised and made the necessary clarifications, this manuscript has my support for publication.

We thank Reviewer #1 again for their time and are glad that they support the publication of our manuscript!

Reviewer #3 (Remarks to the Author):

Good job! The authors have responded to my comments in a careful and considerate manner, their contribution is novel and significant, and the modifications implemented, including for the other reviewers' comments are adequate.

In my opinion, this manuscript is ready for publication.

We thank Reviewer #3 again their time and their kind words. We are glad that they believe our manuscript is ready for publication!